# Assessing knowledge and attitudes toward epilepsy among schoolteachers and students: Implications for inclusion and safety in the educational system

**Luigi Francesco Iannone**[1], **Roberta Roberti**[1], **Gabriele Arena**[1], **Simone Mammone**[1], **Patrizia Pulitano**[2], **Giovambattista De Sarro**[1], **Oriano Mecarelli**[2], **Emilio Russo**[1]*

**1** Science of Health Department, University Magna Graecia, Catanzaro, Italy, **2** Human Neurosciences Department, Sapienza University, Rome, Italy

* erusso@unicz.it

**Data Availability Statement:** All relevant data are within the paper and its Supporting Information files.

## Abstract

Several studies have evidenced inadequate knowledge about epilepsy and inappropriate seizure management, influencing quality of life and social inclusion of patients with epilepsy. Aim of the study was to estimate the knowledge and the attitudes toward epilepsy in schoolteachers and students in Italy. Custom-designed and validated questionnaires in Italian on general and specific knowledge, and social impact of epilepsy have been administered in a random sample of schoolteachers and students. Overall, 667 schoolteachers and 672 students have been included. Among teachers and students, consider epilepsy a psychiatric disorder (16.8% and 26.5%) or an incurable disease (43.9% and 33%). The 47.5% of teachers declared to be unable to manage a seizing student, 55.8% thought it requires specific support and 21.6% reported issues in administer antiseizure medications in school. Healthcare professionals should have an active role in the educational system, dispelling myths, preparing educators and students with appropriate attitudes in the event of a seizure and prevent over limitations in patients with epilepsy. These findings highlight still poor knowledge and attitudes about epilepsy among teachers and students although the 99.4% claimed to have heard/read something about epilepsy. Therefore, improving existing dedicated educational/training interventions could be necessary.

## Introduction

Epilepsy is one of the most common neurological disease that affects about half million people in Italy and has a worldwide prevalence between 5 and 10 *per* 1000 persons [1]. However, despite the scientific advances and the abundance of information available, prejudices, misunderstandings and ignorance persist, carrying a stigma for people with epilepsy that still result in negative social attitudes and discrimination [2, 3].

Several studies have generally demonstrated lack of knowledge about seizures management and negative attitudes in teachers and students, with a deep influence on quality of life and social inclusion of subjects with epilepsy, above all in children attending school [4, 5].

**Funding:** The authors received no specific funding for this work.

**Competing interests:** I have read the journal's policy and the authors of this manuscript have the following competing interests: ER has received speaker fees and participated at advisory boards for Eisai and has received research fundings by GW Pharmaceuticals, Pfizer, Italian Ministry of Health (MoH) and the Italian Medicine Agency (AIFA). All other authors have no conflicts to declare. This does not alter our adherence to PLOS ONE policies on sharing data and materials.

Indeed, schoolchildren with epilepsy have a greater risk of social isolation, poor self-esteem and unnecessary restrictions on involvement in school activities as well as sports, with a significative impact of quality of life (affected more by psychosocial factors than by the seizures themselves) and their future as adults [6, 7].

Teachers' knowledge about and attitudes towards epilepsy have a great and direct influence on students with epilepsy influencing the educational performance, social skills, to correctly frame the disease without needless impairments. Moreover, in the case of a seizure in class, a teacher with basic expertise can be useful for seizure management and to avoid harmful misconducts [8].

Overall, Italian teachers do not receive any formal education about medical disorders, including epilepsy, and numerous studies have demonstrated that teachers have inadequate training, misunderstand the disease and are not able to correctly handle a seizure. Furthermore, teachers, in general, underestimate the academic abilities of students with epilepsy [5].

Nowadays, few studies have been performed in Italy on this topic (*i.e.* one each for students and general population and three for teachers) [9–12] with the same questionnaires and one (*i.e.* investigating both students and teachers) with an *ad hoc* constructed questionnaire [13].

Therefore, we conducted a survey analysis to evaluate and compare the level of knowledge and the attitudes of teachers and students toward epilepsy in Southern Italy, using already validated questionnaires and assessing differences with previous surveys.

## Materials and methods

### Study design and data collection

Custom-designed and validated questionnaires in Italian, partially modified from previous studies [9, 10, 12], on general knowledge, specific knowledge and social impact of epilepsy have been administered in a random sample of primary and secondary schoolteachers as well as upper-middle class and college undergraduate students during 2019. The number of participants enrolled was not defined before the study.

Overall, the questionnaires were structured in 27 and 16 items respectively for teachers and students (only questions regarding attitude are dissimilar between groups) and contained an introductory statement about the purpose and method of the study (S1 Appendix and S2 Appendix). Several questions allowed multiple answers.

The demographics of both teachers and students were also collected and included gender, age, and school of attendance. Years of teaching, specific experience gained whether working with disabled children, participation in training courses on disabling diseases, and number of students with epilepsy was also reported for teachers only. Finally, data have been compared with the first Italian results (performed in 2007 and 2010) with the same questionnaires, to evaluate changes during the last decade. The study protocol was approved by the local Ethics Committee (*Comitato Etico Regionale Calabria*, Italy), protocol number 195/19. Written informed consent was not obtained considering that data were collected and analyzed anonymously.

### Statistical analysis

Descriptive statistical analyses were performed to evaluate demographic characteristics, with continuous data presented as mean ± standard deviation (SD), while ordinal data expressed as number (percentage). The Mann–Whitney U test for continuous variables and the two-tailed Pearson chi-squared test or the Fisher's test for categorical variables have been applied as appropriate. A p-values < 0.05 was considered to indicate statistical significance. Statistical

analysis was performed using the SPSS-26.0 package (IBM Corp. SPSS Statistics, Armonk, NY, USA).

## Results

### Sociodemographic characteristics

Overall, 669 teachers and 680 students have been enrolled. Almost all teachers were female (614, 91.8%), mostly subdivided in the age range 35–54 years and with less than 20 years of teaching (426, 63.7%). To note, almost 90% (600) of teachers have managed in class students with disabilities and the 75% (507) declared to have participated in specific training courses (Table 1).

Furthermore, 46.8% (313) of teachers declared to have a child with epilepsy in their class but just the 50.5% of them was always carefully informed by children' parents to the form of epilepsy (S1 Table in S1 File).

Students had a mean age of 18.4 years, about 70% were female (458) and attended mainly college (500, 73.5%) and in specific, medicine (131, 26.2%) and economics 92 (18.4%) courses (Table 2).

### General and specific knowledge of epilepsy

Overall, two teachers and eight students negatively answered to the first question (i.e. *Do you know the disease called epilepsy*?*)* and have been excluded from the analysis.

Information about epilepsy was mainly acquired through personal/familiar experience for teachers (32.2% vs 14.3, $p < 0.001$) and by hearsay for students (48.7% vs 25.5%, $p < 0.001$), whereas less than 30% in both group attained specific training courses on epilepsy (26.5% vs 22.9%, $p = 0.12$) or used scientific papers (16.6% vs 15.0%, $p = 0.42$).

About one-fourth of teachers (148, 22.2%) and students (114, 17.0%) were aware of the prevalence of epilepsy in Italy, overall considered a less common disease by both groups.

Approximatively 40% of teachers indicate epilepsy as a hereditary or a congenital disorder, with less than 17% thought epilepsy is caused by psychological conditions. On the other hand, students indicate head injuries (42.9%) and psychological disorders (41.1%) as the main cause of epilepsy insurgence. However, only 7.6% (51) of teachers and the 26.5% (178) of students considered epilepsy as a *tout-a-court* psychiatric disease ($p<0.001$).

**Table 1. General characteristics of teacher's sample.**

| Subjects | 669 |
|---|---|
| Sex Male/Female, n (%) | 55 (8.2) / 614 (91.8) |
| Age [y], n (%): | |
| *<35* | 120 (17.9) |
| *35–44* | 227 (33.9) |
| *45–54* | 205 (30.6) |
| *55+* | 117 (17.5) |
| Primary/secondary school, n (%) | 554 (82.8) / 115 (17.2) |
| Teaching [y], n (%): | |
| *<20* | 426 (63.7) |
| *>20* | 243 (36.3) |
| Experience with disabled children, n (%) | 600 (89.7) |
| Attended courses on disability, n (%) | 507 (75.8) |

**Table 2. General characteristics of student's sample.**

| Subjects | 680 |
|---|---|
| Sex Male/Female, n (%) | 222 (32.6) / 458 (67.4) |
| Age [y], mean ± SD | 18.4 ± 0.5 |
| Schoolchildren/ College students | 180 (26.4)/ 500 (73.5) |
| Faculty*, n (%): | |
| Medicine | 131 (26.2) |
| Nursing | 52 (10.4) |
| Economics | 92 (18.4) |
| Humanities | 74 (14.8) |
| Engineering | 72 (14.4) |
| Others | 97 (19.4) |

*For College students only.

Almost the whole teachers' sample (90.4%) knew that the disease can be treated with specific drugs but were not aware of other methods (48, 7.2%) or neurosurgery (92, 13.8%). Accordingly, students mostly indicate specific drugs as treatment option (69.2%) but neurosurgery and other methods were also appropriately indicated (24.9% and 14.7%, respectively). To note, one third of students (33.0%) and the 43.9% of teachers thought that epilepsy is an uncurable illness (p<0.001). Results on general and specific knowledge compared between groups are reported in Table 3.

## Emergency attitudes during a convulsive seizure

Although 52.2% of teachers declare to be able to manage very well or moderately a seizure, the half of both teachers and students would invariably call an ambulance during a convulsive seizure in class (p = 0.46), as well as the 42% and 47% of teachers and students, respectively, thought they should place an object in the subject' mouth (p = 0.03).

To note, almost one third of teachers (190, 28.4%) compared with just the 3% (20) of students, would administer medications endo-rectally (p<0.001) and the 14.4% (117) of students would not know what to do, compared to 4.9% (33) among teachers. All emergency attitudes results are detailed in Table 4.

## School life- and social-related attitudes of teachers

The 21% of respondents declared that the antiseizure medications (ASMs) administration during class was difficult. In general, almost all the teachers manage children with epilepsy in the same way compared to healthy classmates (612, 91.8%) and almost 80% reported normal or positive (i.e. try to help) behaviors of classmates toward children with epilepsy.

Nevertheless, more than half of teachers (372, 55.8%) believed that children with epilepsy require personal support at school, whereas 33.2% reported mental and/or behavior alterations because of epilepsy or treatment with ASMs (276, 41.4%).

Regarding social-related attitude, although the 78.6% of teachers thought that recreational and sports activities should be normal in children with epilepsy, they indicate several sports as not recommended or limited by epilepsy (49.6%). In particular, teachers reported mostly as banned sports: boxing (66.7%), swimming (30.3%), skiing (20.2%) and cycling (19.3%) (S1 Table in S1 File).

Furthermore, teachers reported epilepsy as a limitation to marriage (25.6%), having children (28.3%), regular employment (46.0%) and driving (75.7%) (S2 Table in S1 File).

**Table 3. General and specific knowledge of epilepsy between group.**

| | Teachers (n = 667) | Students (n = 672) | *p* value |
|---|---|---|---|
| *1. Do you know the disease called "epilepsy"?* | | | |
| Yes* | 667 (99.7) | 672 (98.8) | / |
| *2. How do you know epilepsy?* [a]: | | | |
| By hearsay | 170 (25.5) | 327 (48.7) | **<0.0001** |
| Personal or familial experience | 215 (32.2) | 96 (14.3) | **<0.0001** |
| Friends/acquaintances | 96 (14.4) | 116 (17.3) | 0.15 |
| Medical interviews | 109 (16.3) | 45 (6.7) | **<0.0001** |
| Scientific paper | 111 (16.6) | 101 (15.0) | 0.42 |
| Training course | 177 (26.5) | 154 (22.9) | 0.12 |
| *3. Have you ever seen a seizure?* [a] | | | |
| Classroom | 154 (23.1) | 51 (7.6) | **<0.0001** |
| Public place | 143 (21.4) | 115 (17.1) | 0.05 |
| Home | 79 (11.8) | 51 (7.6) | **0.009** |
| TV/movies | 133 (19.9) | 217 (32.3) | **<0.0001** |
| Never | 230 (34.5) | 281 (41.8) | **0.006** |
| *4. What is the approximate prevalence of epilepsy in Italy?* | | | **<0.0001** |
| 1/10 | 11 (1.6) | 25 (3.7) | |
| 1/100 *(correct answer)* | 148 (22.2) | 114 (17.0) | **0.01** |
| 1/1.000 | 258 (38.7) | 203 (30.2) | |
| 1/10.000 | 178 (26.7) | 216 (32.1) | |
| 1/100.000 | 64 (9.6) | 102 (15.2) | |
| 1/1.000.000 | 8 (1.2) | 12 (1.8) | |
| *5. What do you think causes epilepsy?* [a] | | | |
| Hereditary disease | 292 (43.8) | 248 (36.9) | **0.01** |
| Birth defect | 246 (36.9) | 197 (29.3) | **0.003** |
| Viral infection | 114 (17.1) | 117 (17.4) | 0.87 |
| Stress | 45 (6.7) | 103 (20.8) | **<0.0001** |
| Head injury | 232 (34.8) | 288 (42.9) | **0.002** |
| Brain tumor | 198 (29.7) | 213 (31.7) | 0.42 |
| Physiological/psychiatric disease | 112 (16.8) | 276 (41.1) | **<0.0001** |
| *6. What is the age of onset of epilepsy?* | | | **<0.0001** |
| Childhood | 213 (31.9) | 143 (21.3) | |
| Adult | 4 (0.6) | 12 (1.8) | |
| All ages | 349 (59.1) | 434 (64.6) | |
| Do not know | 56 (8.4) | 83 (12.4) | |
| *7. Do you think epilepsy is a form of psychiatric disease?* | | | **<0.0001** |
| Yes | 51 (7.6) | 178 (26.5) | |
| No | 523 (78.4) | 374 (55.7) | |
| Do not know | 93 (13.9) | 120 (17.9) | |
| *8. Do you think epilepsy is treatable with* [a]: | | | **<0.0001** |
| Specific drugs | 603 (90.4) | 465 (69.2) | |
| Neurosurgery | 92 (13.8) | 167 (24.9) | |
| Other methods | 48 (7.2) | 99 (14.7) | |
| Do not know | 43 (6.4) | 119 (7.7) | |
| *9. Do you think epilepsy is a curable illness?* | | | **<0.0001** |
| Yes | 201 (30.1) | 273 (40.6) | |
| No | 293 (43.9) | 222 (33.0) | |

(*Continued*)

**Table 3.** (Continued)

| | Teachers (n = 667) | Students (n = 672) | *p* value |
|---|---|---|---|
| Do not know | 173 (25.9) | 177 (26.3) | |

[a] Multiple answer allowed.

[*]Eight students and two teachers who answered "no" to first question "Do you know the disease called epilepsy?" were excluded.

## Cohorts comparison with previous Italian studies among students and teachers

Among our student's samples, compared with another Italian cohort evaluated in 2007, more students had an overall awareness of epilepsy (98.8% vs 91.0%), with more information obtained through scientific papers and medical advices (15.0% vs 2.0% and 6.7% vs 2.0%, respectively) as well as by hearsay (48.7% vs 37.0%). Less students in our cohort have witnessed a seizure in classroom (7.6% vs 22.0%), whereas the same amount has seen indirectly in common communication media (~33%). No improvements have been achieved in identifying the correct prevalence of epilepsy (approximatively 16% in both cohorts), down estimating the frequency, as well as in the possibility to achieve a permanent seizure remission (~40%).

Notably, compared with previous cohort, less that the half of students considered epilepsy a psychiatric disorder (26.5% vs 56.0%) and were more aware about treatment options (specific drugs 69% vs 59%, neurosurgery 24.9% vs 7%). A considerably higher percentage of students in our sample considered epilepsy an important impediment for driving (87.6% vs 23.0%), sports (58.5% vs 5.0%) and to have children/get married (37.5% vs 12.0%) (S5 Table in S1 File).

As well as students, teachers have been also compared with an Italian cohort that performed the same questionnaire in 2010. Overall, the same percentage of teachers (99.7%) know about epilepsy by information provided evenly by all sources reported in both cohorts (excluding friends/acquaintances). In the 2019, more teachers have witnessed a seizure in TV/movies (19.9% vs 7.5%) whereas no difference has been highlighted in direct experiences (55.3%).

To note, less teachers were aware about the correct prevalence of epilepsy in Italy in our cohort (22.2% vs 33.9%) and were more insecure to identify epilepsy as a psychiatric disease (the 13.9% vs 3.8% did not know answer the question). Furthermore, no differences have been reported in the knowledge of treatment options (specific drugs ~90%, and neurosurgery ~13% in both groups).

Interestingly, the same proportion of teachers persist to identify epilepsy as a limit to marriage, having children and driving, comparing to previous cohort. Furthermore, a higher

**Table 4. Emergency attitude between group.**

| | Teachers (n = 667) | Students (n = 672) | *p* value |
|---|---|---|---|
| *18. In the case of a seizure in class (with loss of consciousness, drop, and spams to the whole body) what would you do?* [a] | | | |
| Call an ambulance | 343 (51.4) | 332 (49.4) | 0.46 |
| Have the person lie down on the ground and wait until the end of the attack | 306 (45.9) | 251 (37.4) | **0.002** |
| Place something in the subject's mouth | 278 (41.6) | 319 (47.5) | **0.03** |
| Block the spasms of the limbs | 45 (6.7) | 131 (19.5) | **<0.0001** |
| Administer medications endo-rectally | 190 (28.4) | 20 (3.0) | **<0.0001** |
| Would not know what to do | 33 (4.9) | 117 (17.4) | **<0.0001** |

[a] Multiple answer allowed.

percentage of teachers in our sample thought that epilepsy is a limitation for a regular employ-ment (56.0% vs 39.7%) and sports (49.8% vs 32.8%). Finally, compared with previous cohort, more teachers thought that children with epilepsy require specific support in school (55.8% vs 36.4%) but were more confident in their skills to manage an epileptic seizure (52.2% vs 33.6%).

## Discussion

Epilepsy can significantly affect the quality of life of patients, above all during childhood, where school (children spend about 40% of their time at school) or other community settings, have a critical role to educate and avoid discrimination, that may influence academic and per-sonal achievements during lifetime. Teachers with correct knowledge about epilepsy and appropriate attitudes can deeply change the prejudices and stigma surrounding this disease. Further and foremost, the appropriate rescue attitudes by teachers and students, during pro-longed acute convulsive seizures, can be decisive to avoid delay in treatment, injuries, and unnecessary use of emergency services.

This survey has shown that teachers and students are fully aware of the existence of epilepsy and having experience (mostly directly for teachers and indirectly for students) of the disease. Nevertheless, specific knowledge and attitudes (above all during a seizure) are lacking or insuf-ficient and harmful (approximatively the 40% of students and teachers thought they should place an object in the subject' mouth during a seizure).

Although 50% of teachers declare to know how to manage a seizure, half of the interviewed would call invariably an ambulance, independently of duration, type, and other subject's fea-tures, as recommended by guidelines [8, 14]. Regarding teachers, they reported a diffused mis-communication with parents, with half of them not always informed about students' conditions. Several guidelines have been published in European countries, but the actual appli-cability is high variable across countries and schools, sometimes limited by concerns on legal implications.

However, results are in line with the majority of previous studies in Italy [12, 13, 15] and worldwide [16–20] or slightly better compared to others [9, 21].

In the last decade, Italian students improved the awareness of epilepsy (98.8% vs 91.0%), with more information obtained through scientific articles and medical advices. Furthermore, students and teachers that indicate epilepsy as a psychiatric disorder were largely reduced (-30% and -3.3%, respectively) and were more aware about treatment options. This could be related to the promotion of several nationwide campaigns by the Italian League Against Epi-lepsy (LICE) and other associations to improve public knowledge on epilepsy [15]. The success of these educational programs was already demonstrated in a two stages (before and after) cross-sectional study in Italy [12].

However, a largely increased number of students and teachers believe that epilepsy provide social disadvantage, with limitation for driving, marriage, sports, and other activities. Further-more, teachers continue to indicate epilepsy (or ASMs) as a source of learning disability that require specific support in school. These findings are strongly related with prejudice and, as reported by Mecarelli et al., education is more likely to improve ignorance that preconceptions [15].

Although our samples of students and teachers have been enrolled in a limited geographic area, they seem representative of the overall Italian territory. Indeed, teachers' features are in line with previous studies and in agreement to data of the *Organization for the Economic Coop-eration and Development* [22], whereas students belong to different faculties as well as second-ary school.

Our study had some limitations. We could not accurately identify all the factors that influence knowledge and attitudes toward epilepsy and the questions setting is limited, investigating selected areas. Finally, the selection of school to be investigated may have resulted in a response bias, as well as the voluntary participation to the study.

## Conclusion

The experiences of students and teachers reported in this paper underline that epilepsy continues to be persistently mismanaged in the educational system, impacting on patients with epilepsy due to inappropriate limitations and epilepsy-related stigma. Furthermore, epilepsy is a highly sensitive topics for families and parents need to be cognizant of broaching of epilepsy stigma and hence improve (or create) communication strategies with school and other institutions.

Eliminate preconceptions and outline specific and effective programs to train teachers and students to manage a seizure are persistent challenges for healthcare professionals. Indeed, parents and teachers might necessitate assistance [23] in learning how to initiate and maintain an active collaboration, as well as to discriminate accurate and helpful information to myths or erroneous advices.

Our findings suggest an improved knowledge and social awareness about epilepsy during the last decade but persistent unacceptable attitudes. Therefore, healthcare communication should have an active role in the educational system, dispelling myths, preparing teachers and students with appropriate attitudes in case of seizures and prevent over limitations in patients with epilepsy.

Future programs are needed to increase communicative interventions to eradicate the stigma as well as to improve attitudes about epilepsy and should not be postponed.

## Supporting information

**S1 File.**
(DOCX)

**S1 Appendix. Teachers survey questionnaire.**
(DOCX)

**S2 Appendix. Students survey questionnaire.**
(DOCX)

## Author Contributions

**Conceptualization:** Oriano Mecarelli, Emilio Russo.

**Data curation:** Luigi Francesco Iannone, Roberta Roberti, Gabriele Arena, Simone Mammone.

**Formal analysis:** Luigi Francesco Iannone.

**Investigation:** Simone Mammone.

**Methodology:** Oriano Mecarelli.

**Project administration:** Emilio Russo.

**Resources:** Luigi Francesco Iannone, Gabriele Arena.

**Supervision:** Patrizia Pulitano, Giovambattista De Sarro, Oriano Mecarelli, Emilio Russo.

**Validation:** Patrizia Pulitano, Giovambattista De Sarro, Emilio Russo.

**Writing – original draft:** Luigi Francesco Iannone.

**Writing – review & editing:** Luigi Francesco Iannone, Roberta Roberti, Patrizia Pulitano, Giovambattista De Sarro, Oriano Mecarelli, Emilio Russo.

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
