## [Decision Letter · Decision Letter 0]

12 Mar 2021

PONE-D-20-38678

Assessing knowledge and attitudes toward epilepsy among schoolteachers and students: implications for inclusion and safety in the educational system

PLOS ONE

Dear Dr. Russo,

Thank you for submitting your manuscript to PLOS ONE. After careful consideration, we feel that it has merit but does not fully meet PLOS ONE’s publication criteria as it currently stands. Therefore, we invite you to submit a revised version of the manuscript that addresses the points raised during the review process.

Please provide replies to minor comments

The authors could briefly discuss similar studies in other countries if any.

We look forward to receiving your revised manuscript.

Kind regards,

Andrea Romigi, M.D., Ph.D

Academic Editor

PLOS ONE

Journal Requirements:

3. Please provide additional details regarding participant consent. In the ethics statement in the Methods and online submission information, please ensure that you have specified (1) whether consent was informed and (2) what type you obtained (for instance, written or verbal, and if verbal, how it was documented and witnessed). If your study included minors, state whether you obtained consent from parents or guardians. If the need for consent was waived by the ethics committee, please include this information."

4. Please include additional information regarding the survey or questionnaire used in the study and ensure that you have provided sufficient details that others could replicate the analyses. For instance, if you developed a questionnaire as part of this study and it is not under a copyright more restrictive than CC-BY, please include a copy, in both the original language and English, as Supporting Information.  If the original language is written in non-Latin characters, for example Amharic, Chinese, or Korean, please use a file format that ensures these characters are visible.

5. Please state whether you validated the questionnaire prior to testing on study participants. Please provide details regarding the validation group within the methods section.

Reviewers' comments:

Reviewer's Responses to Questions

**Comments to the Author**

1. Is the manuscript technically sound, and do the data support the conclusions?

Reviewer #1: Yes

Reviewer #2: Yes

2. Has the statistical analysis been performed appropriately and rigorously? 

Reviewer #1: Yes

Reviewer #2: N/A

3. Have the authors made all data underlying the findings in their manuscript fully available?

Reviewer #1: No

Reviewer #2: Yes

4. Is the manuscript presented in an intelligible fashion and written in standard English?

Reviewer #1: Yes

Reviewer #2: Yes

5. Review Comments to the Author

Reviewer #1: This was a survey-based study aimed to estimate the knowledge and attitudes toward epilepsy among Italian schoolteachers and students in Italy. Overall, 667 schoolteachers and 672 students were included.

The study is very interesting and has practical implication by highlighting the need to improve the general knowledge about seizures and epilepsy in lay population. There are, however, some issues that could be better addressed.

In the abstract, the phrase “Among teachers and students, consider epilepsy a psychiatric disorder (16.8% and 26.5%) or an incurable disease (43.9% and 33%)” should be reworded for greater clarity.

The comparison of the result of the current study with the findings from previous similar reports should be better placed in the discussion rather than result section.

Reviewer #2: This study estimated the knowledge and the attitudes toward epilepsy in schoolteachers and students in Italy. Custom-designed and validated questionnaires on knowledge and social impact of epilepsy have been administered in a random sample of teachers and students. Overall, the questionnaires were structured in 27 and 16 items respectively for teachers and students (only questions regarding attitude are dissimilar between groups) and contained an introductory statement about the purpose and method of the study (Appendix A and B). Several questions allowed multiple answers. Overall, 667 teachers and 672 students have been included. Among teachers and students, consider epilepsy a psychiatric disorder (16.8% and 26.5%) or an incurable disease (43.9% and 33%). The 47.5% of teachers declared to be unable to manage a seizing student, 55.8% thought it requires specific support and 21.6% reported issues in administering medications in school. Healthcare professionals should have an active role in the educational system, dispelling myths, preparing educators and students with appropriate attitudes in the event of a seizure, and prevent over limitations in patients with epilepsy.

Comment

Several studies have evidenced inadequate knowledge about epilepsy and inappropriate seizure management, influencing the quality of life and social inclusion of patients with epilepsy. The experiences of students and teachers reported in this paper underline that epilepsy continues to be persistently mismanaged in the educational system, impacting patients with epilepsy due to inappropriate limitations and epilepsy-related stigma. Therefore, these findings highlight still poor knowledge and attitudes about epilepsy among teachers and students. Improving existing dedicated educational/training interventions are needed.

The manuscript is well written and statistical analysis is correctly addressed. The authors could briefly discuss similar studies in other countries if any.

6. PLOS authors have the option to publish the peer review history of their article (what does this mean?). If published, this will include your full peer review and any attached files.

Reviewer #1: No

Reviewer #2: **Yes: **Pasquale Striano

---

## [Author Response · Author response to Decision Letter 0]

17 Mar 2021

Reviewers Comments to the Author

Reviewer #1: This was a survey-based study aimed to estimate the knowledge and attitudes toward epilepsy among Italian schoolteachers and students in Italy. Overall, 667 schoolteachers and 672 students were included. The study is very interesting and has practical implication by highlighting the need to improve the general knowledge about seizures and epilepsy in lay population. There are, however, some issues that could be better addressed.

In the abstract, the phrase “Among teachers and students, consider epilepsy a psychiatric disorder (16.8% and 26.5%) or an incurable disease (43.9% and 33%)” should be reworded for greater clarity.

The comparison of the result of the current study with the findings from previous similar reports should be better placed in the discussion rather than result section.

The sentence in the abstract has been reworded accordingly: “Moreover, some students and teachers believe that epilepsy is a psychiatric disorder (16.8% and 26.5%) or an incurable disease (43.9% and 33%), respectively.”

The comparison of the results with previous reports has been reported in the discussion ‘section.

Reviewer #2: 

Comment

Several studies have evidenced inadequate knowledge about epilepsy and inappropriate seizure management, influencing the quality of life and social inclusion of patients with epilepsy. The experiences of students and teachers reported in this paper underline that epilepsy continues to be persistently mismanaged in the educational system, impacting patients with epilepsy due to inappropriate limitations and epilepsy-related stigma. Therefore, these findings highlight still poor knowledge and attitudes about epilepsy among teachers and students. Improving existing dedicated educational/training interventions are needed.

The manuscript is well written and statistical analysis is correctly addressed. The authors could briefly discuss similar studies in other countries if any.

Difference between out study and other study worldwide have been reported in the discussion section:

“However, results are in line with the majority of previous studies in Italy [12,13,15] and worldwide [16–20] or slightly better compared to others [9,21]. In the last decade, several studies demonstrated an improved positive attitude during the years despite the continuous lack of knowledge about epilepsy and numerous variables have been implicated, including have a relative with epilepsy or previous experience with a student with epilepsy for teachers [17][22]. Overall, improved knowledge about the disease seem to positively affects attitude toward a person with epilepsy [23], although social preconceptions persist.”

---

## [Editor Report · Decision Letter 1]

23 Mar 2021

Assessing knowledge and attitudes toward epilepsy among schoolteachers and students: implications for inclusion and safety in the educational system

PONE-D-20-38678R1

Dear Dr. Russo,

We’re pleased to inform you that your manuscript has been judged scientifically suitable for publication and will be formally accepted for publication once it meets all outstanding technical requirements.

Kind regards,

Andrea Romigi, M.D., Ph.D

Academic Editor

PLOS ONE
---

## [Editor Report · Acceptance letter]

25 Mar 2021

PONE-D-20-38678R1 

Assessing knowledge and attitudes toward epilepsy among schoolteachers and students: implications for inclusion and safety in the educational system 

Dear Dr. Russo:

I'm pleased to inform you that your manuscript has been deemed suitable for publication in PLOS ONE. Congratulations! Your manuscript is now with our production department. 

Kind regards, 

on behalf of

Dr. Andrea Romigi 

Academic Editor

PLOS ONE